# Are Cervical Length and Fibronectin Predictors of Preterm Birth after Fetal Spina Bifida Repair? A Single Center Cohort Study

**DOI:** 10.3390/jcm12010123

**Published:** 2022-12-23

**Authors:** Ladina Vonzun, Ladina Rüegg, Julia Zepf, Ueli Moehrlen, Martin Meuli, Nicole Ochsenbein-Kölble

**Affiliations:** 1Department of Obstetrics, University Hospital Zurich, Rämistrasse 100, 8091 Zurich, Switzerland; 2Faculty of Medicine, University of Zurich, Rämistrasse 71, 8091 Zurich, Switzerland; 3The Zurich Center for Fetal Diagnosis and Therapy, University of Zurich, 8091 Zurich, Switzerland; 4Spina Bifida Academy, University Children‘s Hospital Zurich, Steinwiesstrasse 75, 8032 Zurich, Switzerland; 5Spina Bifida Center, University Children‘s Hospital Zurich, Steinwiesstrasse 75, 8032 Zurich, Switzerland; 6Department of Pediatric Surgery, University Children‘s Hospital Zurich, Steinwiesstrasse 75, 8032 Zurich, Switzerland

**Keywords:** fetal surgery, spina bifida, cervical length, fibronectin test, preterm birth, PPROM

## Abstract

Background: A remaining risk of fetal spina bifida (fSB) repair is preterm delivery. This study assessed the value of preoperative cervical length (CL), CL dynamics (∆CL) and fetal fibronectin (fFN) tests to predict obstetric complications and length of stay (LOS) around fSB repair. Methods: 134 patients were included in this study. All patients had CL measurement and fFN testing before fSB repair. ∆CL within the first 14 days after intervention and until discharge after fSB repair were compared in groups (∆CL ≥ 10 mm/<10 mm; ≥20 mm/<20 mm). CL before surgery, ∆CL’s, and positive fFN tests were correlated to obstetric complications and LOS. Results: Mean CL before surgery was 41 ± 7 mm. Mean GA at birth was 35.4 ± 2.2 weeks. In the group of ∆CL ≥ 10 mm within the first 14 days after intervention, LOS was significantly longer (*p* = 0.02). ∆CL ≥ 10 mm until discharge after fSB was associated with a significantly higher rate of GA at birth <34 weeks (*p* = 0.03). The 3 positive fFN tests before fSB repair showed no correlation with GA at birth. Conclusion: Perioperative ∆CL influences LOS after fetal surgery. ∆CL ≥ 10 mm until discharge after fSB repair has a 3-times higher rate of preterm delivery before 34 weeks. Preoperative fFN testing showed no predictive value for preterm birth after fSB repair and was stopped.

## 1. Introduction

About 1 in 3000 pregnancies is affected by spina bifida aperta (SB), which is the most common neural tube defect [1,2]. Nowadays, the intrauterine repair of a SB is a valid therapeutic option for selected cases [3,4,5]. It has been shown to reduce the need for ventriculoperitoneal shunt placement and to improve long-term neurological function [3,5]. Even though fetal spina bifida (fSB) repair is a standard therapy in several centers, especially in the US and Europe, there are risks for mother and fetus to be considered [6,7]. Intrauterine surgery is associated with obstetrical complications such as preterm prelabour rupture of the membranes (PPROM), chorioamniotic membrane separation (CMS), and subsequently preterm birth with its adverse consequences for the newborn [7,8,9,10,11]. Preterm birth is, independently from fSB repair, the most frequent cause of perinatal mortality and severe perinatal morbidity in the Western world [12].

A short cervical length (CL) is known to be a good predictor of preterm birth [13,14]. Especially in high-risk pregnancies, measuring CL is valuable and helpful for further management [15]. For fetal interventions, such as laser therapies in twin pregnancies, CL measurement before surgery is the most important predictor for preterm birth [16].

A further potential marker for preterm delivery is the fetal fibronectin (fFN) test [17]. fFN is a glycoprotein found in placental tissue, the extracellular substance of the decidua, and in the amniotic fluid. Through inflammatory or mechanical damage to the membranes before birth, it is thought to be released. A fFN test showing positive results indicates a high likelihood of preterm delivery within the next 7–10 days [18]. In the context of fSB repair, the value of CL measurements or fFN tests for the risk assessment of complications has never been evaluated.

Therefore, the goal of this study was to assess whether the preoperative CL measurements and its dynamics (∆CL) and/or a positive fFN test before surgery is associated with preterm birth, PPROM, or CMS. An additional aim was to evaluate whether short CL measurements have an impact on the length of stay (LOS) after fSB repair.

## 2. Materials and Methods

### 2.1. Patients and Study Design

Between December 2010 and April 2020, 136 women underwent fSB repair at the Zurich Center for Fetal Diagnosis and Therapy (www.swissfetus.ch, accessed on 20 November 2022). Two patients were excluded from this study due to withdrawn consent. Data were collected prospectively and entered into our RedCap database. If available, missing data was completed by retrospective analysis of the ultrasound reports and images.

Eligibility criteria, the peri- and postoperative management, and the open surgical technique have been published previously in detail [4,5,7,19,20].

All patients received CL measurement before surgery. Postoperative CL measurements performed within the first 14 days after fetal surgery were considered for perioperative ∆CL assessment. CL measurements before discharge were considered for the assessment of CL dynamic until discharge after fSB repair. Analysis of ∆CL within the first 14 days after fetal surgery and until the end of hospitalization were performed in subgroups (∆C ≥ 10 mm or <10 mm and ∆C ≥ 20 mm or <20 mm). Cut-offs were chosen in 10 mm steps as smaller changes likely underlie higher interobserver variability. All findings were correlated to the GA at birth, obstetric complications such as CMS and PPROM, and LOS. Ultrasound examination was performed by one of our senior consultants, specialist in prenatal ultrasound. A GE Voluson E8 or E10 system (GE Healthcare Austria GmbH & CO OG, Zipf, Austria) transvaginal transducer was used. CL was measured longitudinally, from the inner to the outer uterine orifice taking into account the natural curve of the cervix, and with an empty bladder, as recommended by the ISUOG Guidelines [21]. In case of preoperative CL below the 5th percentile according to the nomogram by Papastefanou et al. [22], a pessary (Arabin^®^ Cerclage Pessar) was placed during fetal surgery. After placement of a pessary, no more routine CL measurements took place throughout the further course of pregnancy.

Furthermore, fFN test (QuickCheck fFN, Hologic Inc., Marlborough, MA, USA) was performed routinely before surgery. Swabs were taken from the ectocervix, or the posterior vaginal fornix and an enzyme linked immunosorbent assay (ELISA) with FDC6 monoclonal antibody was used to detect. The perioperative management did not differ in women with positive or negative fFN tests.

All patients received perioperative tocolysis as described previously [23]. After fSB repair, all women were monitored in an intensive care unit for 2 days where contractions were continuously monitored by tocography (IntelliSpace, Perinatal information system, Philips AG Healthcare, Horgen, Switzerland). Ultrasound was performed twice daily to check for amniotic fluid, CMS, hematoma formation, and fetal perfusion. After transferring the patient to our prenatal unit, contractions were monitored by tocography twice a day, and ultrasound was performed once a week. This regimen has been previously published in detail [23]. If clinically stable and in the absence of complications such as, e.g., PPROM, discharge was possible 2–3 weeks after fSB repair with planned preventative re-hospitalization at 34 weeks and elective C-section at 37 weeks.

### 2.2. Data Analysis

Descriptive statistics were performed with SPSS version 25.0 (IBM, SPSS Inc., Armonk, NY, USA). Quantitative data are presented as mean +/− standard deviation (SD) or median with interquartile range (IQR) depending on the distribution of the data. Categorical data were compared using chi square test and provided as percentages. For comparison of continuous data, a *t*-test or non-parametric analysis using Mann–Whitney U test was performed as appropriate. The Pearson correlations coefficient was used to measure the strength of relationship between variables. Statistical significance was given with *p* < 0.05.

This study was conducted with the principles of the Declaration of Helsinki and International Conference on Harmonisation E6 (Good Clinical Practice) guidelines. Written informed consent was obtained from all included women. The study was conducted in accordance with the approval of the local Ethic Commission (KEK-ZH. Nr. 2021-01101).

## 3. Results

Detailed patient’s characteristics are shown in Table 1. The mean gestational age (GA) at birth was 35.4 ± 2.2 weeks with a total preterm birth rate of 66% and 20% before 34 weeks.

### 3.1. Cervical Length (CL) Measurements

An overview on perioperative CL measurements and fFN test results is given in Table 2.

CL measurements before and after fSB repair are shown in Figure 1 on the nomogram by Papastefanou et al. [22].

Mean CL before surgery was 41 ± 7 mm. In 88 (65%) cases, the CL was ≥40 mm. Two (1.5%) women had a preoperative CL < 25 mm, of which both were below the 5th percentile, and thus a pessary was placed during fetal intervention.

In two further cases (1.5%) postoperative CL measurement was below the 5th percentile with a dynamic of 18 mm in one and 10 mm in the other case. Consequently, a pessary was placed upon diagnosis. In these four women presenting CL measurements below the 5th percentile, the LOS was comparable to the other cases (21 (18–23) vs. 23 (19–38)) (*p* = 0.7) and they all delivered between 35 and 37 weeks.

There was no significant correlation between the CL before surgery and GA at birth (*p* = 0.83). Preoperative CL measurements showed no significant correlation with the obstetrical complications as CMS (*p* = 0.28) and PPROM (*p* = 1.0), but a trend was seen for the LOS (*p* = 0.06) showing a slightly longer hospitalization when CL was shorter.

Further, there was no significant correlation between the overall ∆CL within 14 days after the intervention nor until the end of hospitalization and GA at delivery, CMS, PPROM and duration of hospitalization (Table 3).

Subgroup analysis of the perioperative results with the ones of the first 14 days after fetal intervention showed a significant difference in GA at delivery for ∆CL ≥ 20 mm (*n* = 2) (Table 4). No significant correlation with the analyzed obstetrical complications CMS and PPROM were observed within the subgroups of ∆CL ≥ 20 mm or <20 mm. In the group of perioperative ∆CL ≥ 10 mm, a significantly shorter hospitalization was observed (*p* = 0.02) (Table 4).

Analysis of ∆CL until the end of hospitalization after fSB repair showed a significantly (3-times) higher rate of preterm birth before 34 weeks (*p* = 0.03), if ∆CL was ≥10 mm (*n* = 9) in comparison to the group of ∆CL < 10 mm (*n* = 3) (Table 5). In the group with ∆CL ≥ 10 mm, the mean GA at delivery was still 35.1 weeks and the rate of extremely and very preterm born infants was not significantly different between the two groups (Table 5). In the group of ∆CL > 10 mm a nearly 4-times higher rate of CMS was found (*p* = 0.045) (Table 5). Using the higher cut-off of ∆CL ≥ 20 mm, these differences could not be reproduced (Table 5).

### 3.2. Fetal Fibronectin (fFN) Test

Out of the 134 patients, fFN testing was performed in 121 (89%) cases. It was positive in 3 (2%) of these cases (GA 24–25 weeks) (Table 2). In all 3 cases, the preoperative CL was >35 mm (37–51 mm). Two of these women delivered at 37 weeks, one at 35 weeks, thus not showing an obvious correlation between the positive fFN test and GA at birth. Due to the low numbers, no statistical analysis was performed.

## 4. Discussion

CMS, PPROM, and ultimately preterm delivery are known risks after fetal surgery. In our study cohort, neither preoperative CL measurements nor perioperative ∆CL showed any correlation with preterm birth, CMS, or PPROM. However, perioperative ∆CL seems to influence LOS and a ∆CL ≥ 10 mm until the end of hospitalization is associated with a 3-times higher rate of preterm delivery before 34 weeks.

### 4.1. Cervical Length and Preterm Delivery

Independent from fetal surgery, CL is a known predictor for preterm delivery in asymptomatic women at increased risk for preterm birth [24]. Rottenstreich et al. [25] observed in a case–control study that a CL dynamic of ≥4 mm between 24 and 34 weeks had a higher rate of preterm deliveries compared to women matched for the maximal cervical length (<35 weeks: 15.9 vs. 5.3%, *p* = 0.013). The rates were comparable to women matched according to the minimal CL [25]. We did not find a difference in rate of preterm deliveries, CMS, or PPROM with CL dynamics.

For fetal interventions, such as laser therapies in twin pregnancies, a CL < 25 mm before surgery is associated with a lower GA at birth and worse outcomes at 6 months of age [16]. It therefore plays a crucial role in selecting candidates and for parental counseling [16]. Other studies on fSB repair even excluded patients with a high risk for preterm delivery (previous preterm delivery or short CL) from eligibility [26]. In light of the above considerations, it seemed logical that CL measurements before, or even more, CL dynamics around fetal surgery, might influence preterm birth rates. Thus, in our cohort, a high perioperative ∆CL of ≥20 mm showed a five-time higher preterm birth rate <34 weeks, very close to statistical significance (*p* = 0.054). However, we found that a cut-off of ∆CL ≥ 10 mm until the end of hospitalization was significantly associated with preterm birth before 34 weeks (*p* = 0.03), but not with a higher rate of extreme or very preterm births.

Our results (open repair!) are also in line with the results of an open fSB repair cohort published by Da Rocha et al. [27] who observed that CL before surgery was comparable between women who delivered before 34 (*n* = 22) and after 34 gestational weeks (*n* = 17). He further described a PPROM rate of 46% and a rate of preterm delivery of 77%, with chorioamnionitis being the only associated risk factor for preterm delivery and/or PPROM. Their mean GA at delivery was 33.2 ± 3.7 GW [27].

Several studies report their numbers of preterm deliveries, CMS, and PPROM rates [26,28,29]. CL measurements, however, were rarely taken into account. Retrospectively, we assume our strict pre- and postoperative management, as well as our rigorous rule of ‘zero tolerance for contractions’ might have prevented an association between short cervix and premature birth. Additionally, women with a short CL (<5th P) pre- or postoperatively received a cervical pessary possibly contributing to fewer preterm deliveries.

Taken together our results and the above considerations, independent factors such as CMS, PPROM and contractions definitely seem to play a more important role in prediction of preterm birth than preoperative CL measurements or perioperative ΔCL. The authors therefore conclude that neither preoperative CL measurements nor perioperative ΔCL, in the absence of other symptoms and unless CL measurement is <5th P, must result in a different care management of patients that undergo or underwent fSB repair.

### 4.2. Fetal Fibronectin (fFN) Testing

The prediction of preterm birth using fFN tests remains controversial. Honest et al. [18] published a systematic review and found impending that fFN testing is most accurate in predicting spontaneous preterm birth within the next 7–10 days in symptomatic women [18]. On the other hand, a systematic review and metaanalysis of randomized controlled trials from Berghella et al. [30] showed that fFN testing in singleton pregnancies with imminent or impending preterm labor is associated with higher cost, but does not prevent preterm birth nor improve in perinatal outcome [30].

Pinheiro et al. [31] performed a risk assessment for preterm delivery using fFN and CL measurement in symptomatic women. They concluded that a positive fFN and CL < 25 mm indicate an increased risk for preterm delivery within the next 14 days. In our cohort, we did not have any such cases.

Interestingly, a study by Chon et al. [32] looked at the quantitative fetal fibronectin (qfFN) and its association with spontaneous preterm birth after laser surgery for twin-to-twin transfusion syndrome. They collected qfFN 24 h before and after surgery. The qfFN level was not altered after surgery in their cohort. However, they reported that patients with a qfFN levels > 10 ng/mL were 19 times more likely to have spontaneous preterm delivery before 28 GW. During open fSB repair manipulation on fetal membranes is inevitably stronger. Assumed to be less sensitive, postoperative FN testing was therefore not in our protocol and thus not evaluated in our collective. Yet, our study could not detect women at risk for preterm birth after fSB by preoperative fFN testing. Consequently, considering cost-effectiveness its use was stopped.

### 4.3. Strengths and Weaknesses

Positively, to our knowledge, this is the first study on a large cohort to examine preoperative CL measurements and different ∆CL around fSB repair. We show that preoperative CL measurements and short-term perioperative ∆CL do not influence GA at birth. However, long-term ∆CL of ≥10 mm may help predicting preterm delivery <34 weeks. Additionally, this is the first study to examine preoperative fFN tests in the context of fetal surgery.

Negatively, the numbers of women with short CL and positive fFN tests were too small to allow statistical analyses. No qfFN analysis was made and consequently, since we did not perform postoperative fFN testing, no statement regarding an eventual postoperative predictive validity can be made.

Further, in cases with CL below the 5th percentile, a prompt pessary treatment was installed, potentially leading to a positive bias concerning the correlation between shortened CL and preterm birth.

## 5. Conclusions

Perioperative ∆CL seems to influence the LOS after fetal surgery. Further, a ∆CL ≥ 10 mm until discharge after fSB repair has a 3-times higher rate of preterm delivery before 34 weeks.

Preoperative fFN testing showed no predictive value for preterm birth after fSB repair and was consequently stopped.

## Figures and Tables

**Figure 1 jcm-12-00123-f001:**
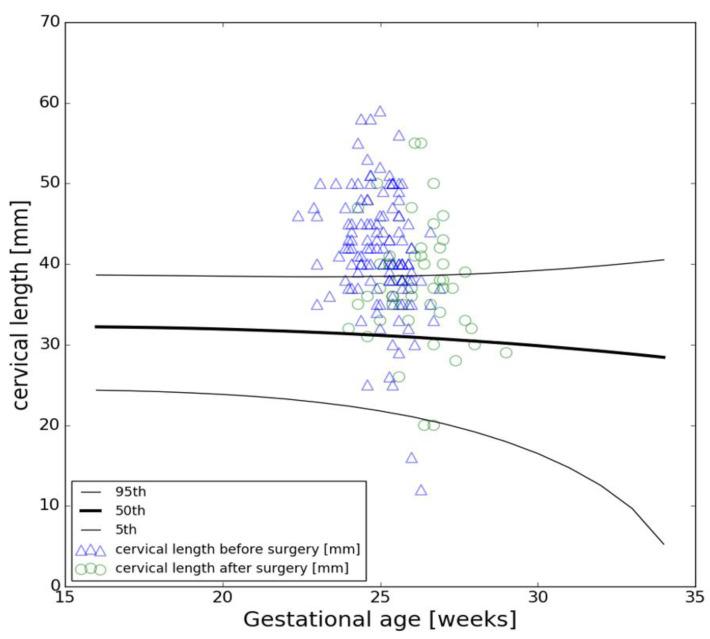
Overview of cervical length measurements before and after surgery on the nomogram by Papastefanou et al. [22].

**Table 1 jcm-12-00123-t001:** Maternal and fetal characteristics of the Zurich study cohort.

Demographics	*n* = 134
Maternal age (years), mean ± SD	31.9 ± 5
Nulliparous, no. (%)	58 (43)
Body mass index (kg/m^2^), mean ± SD	25.8 ± 4.8
Previous uterine surgeries, incl. cesarean, no. (%)	21 (16)
GA at surgery (weeks), mean ± SD	25.0 ± 0.8
GA at birth (weeks), mean ± SD	35.4 ± 2.2
extreme preterm (≤28 0/7 weeks), no. (%)	2 (1.5%)
very preterm (28 0/7–31 6/7 weeks), no. (%)	8 (5.9%)
moderately preterm (32 0/7–33 6/7 weeks), no. (%)	17 (12.5%)
late preterm (34 0/7–36 6/7 weeks), no. (%)	61 (45.5%)
term (≥37 0/7 weeks), no. (%)	47 (34.5%)
Birthweight (grams), mean ± SD	2565 ± 518
PPROM, no. (%)	44 (33%)
CMS, no. (%)	22 (16%)
LOS (days), median (IQR)	23 (19–38)

**Table 2 jcm-12-00123-t002:** Results of cervical length (CL) before surgery and the fetal Fibronectin tests (fFN).

Evaluation before Surgery	*n* = 134
CL before surgery (mm), mean ± SD	41 ± 7
<25 mm, no. (%)	2 (1.5%)
≥25 mm, no. (%)	132 (99%)
fFN	
Positive, no (%)	3 (2%)
Negative, no. (%)	116 (87%)
No test results, no. (%)	15 (11%)

**Table 3 jcm-12-00123-t003:** Overview of the missing correlation between the overall cervical length dynamics (∆CL) within 14 days after fetal spina bifida repair and the ∆CL until the end of hospitalization, respectively, and GA at birth, chorioamniotic membrane separation (CMS), premature preterm prelabour rupture of the membranes (PPROM), and length of stay (LOS).

	∆CL within 14 d		∆CL until End of Hospitalisation	
	corr.coeff.	*p*-Value	corr.coeff.	*p*-Value
GA at birth	0.04	0.79	0.02	0.86
CMS	−0.12	0.42	−0.2	0.09
PPROM	−0.06	0.69	0.15	0.23
LOS	0.15	0.24	0.16	0.21

**Table 4 jcm-12-00123-t004:** Comparison of the groups with cervical length dynamics (∆CL) ≥10 mm or <10 mm and ≥20 mm or <20 mm within the first 14 days after fetal spina bifida repair.

	∆CL ≥ 10 mm (*n* = 10)	∆CL < 10 mm (*n* = 40)	*p*-Value	∆CL ≥ 20 mm (*n* = 2)	∆CL < 20 mm (*n* = 48)	*p*-Value
CL preoperative, mm (mean ± SD)	46 ± 9	40 ± 6	0.03	57 ± 2	40 ± 7	0.001
GA at delivery, weeks (mean ± SD)	35.2 ± 2.3	35.1 ± 2.6	0.9	32.0 ± 0.5	35.3 ± 2.5	0.03
GA at delivery, *n* (%) <28 weeks 28 + 0–31 + 6 weeks 32 + 0–33 + 6 weeks 34 + 0–36 + 6 weeks ≥37 + 0 weeks	0 (0%) 1 (10%) 2 (20%) 3 (30%) 4 (40%)	1 (2.5%) 5 (12.5%) 3 (7.5%) 19 (47.5%) 12 (30%)	0.8 0.66 0.26 0.26 0.40	0 (0%) 1 (50%) 1 (50%) 0 (0%) 0 (0%)	1 (2%) 5 (10%) 4 (8%) 22 (46%) 16 (34%)	0.96 0.23 0.19 0.31 0.46
GA at delivery < 34 weeks, *n* (%)	3 (30%)	9 (23%)	0.45	2 (100%)	10 (21%)	0.05
CMS, *n* (%)	1 (10%)	7 (18%)	0.45	1 (50%)	7 (15%)	0.3
PPROM, *n* (%)	3 (30%)	13 (33%)	0.6	1 (50%)	15 (31%)	0.54
LOS, days (median, IQR)	19 (17–24)	30 (21–40)	0.02	22 (19–23)	28 (20–39)	0.49

**Table 5 jcm-12-00123-t005:** Comparison of the groups with cervical length dynamics (∆CL) ≥10 mm or <10 mm and ≥20 mm or <20 mm until discharge after fetal spina bifida repair.

	∆CL ≥ 10 mm (*n* = 30)	∆CL < 10 mm (*n* = 34)	*p*-Value	∆CL ≥ 20 mm (*n* = 11)	∆CL < 20 mm (*n* = 53)	*p*-Value
CL preoperative, mm (mean ± SD)	44 ± 10	40 ± 5	0.02	47 ± 12	41 ± 7	0.02
GA at delivery, weeks (mean ± SD)	35.1 ± 2.7	36.0 ± 1.8	0.12	35.3 ± 2.1	35.6 ± 2.3	0.35
GA at delivery, *n* (%) <28 weeks 28 + 0–31 + 6 weeks 32 + 0–33 + 6 weeks 34 + 0–36 + 6 weeks ≥37 + 0 weeks	1 (3%) 3 (10%) 5 (17%) 13 (43%) 8 (27%)	0 (0%) 1 (3%) 2 (6%) 17 (48%) 15 (43%)	0.46 0.33 0.23 0.67 0.17	0 (0%) 2 (18%) 1 (9%) 5 (46%) 3 (27%)	1 (2%) 2 (4%) 6 (11%) 24 (45%) 20 (38%)	0.82 0.14 0.64 0.62 0.4
GA at delivery < 34 weeks, *n* (%)	9 (30%)	3 (9%)	0.03	3 (27%)	9 (17%)	0.36
CMS, *n* (%)	7 (23%)	2 (6%)	0.05	2 (18%)	7 (13%)	0.5
PPROM, *n* (%)	9 (30%)	6 (17%)	0.22	4 (36%)	11 (21%)	0.25
LOS, days (media, IQR)	30 (19–40)	26 (20–38)	1.0	29 (19–36)	31 (18–39)	0.86

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
