# Peer review of "Are Cervical Length and Fibronectin Predictors of Preterm Birth after Fetal Spina Bifida Repair? A Single Center Cohort Study"

_jcm, 2022, doi:10.3390/jcm12010123_

Round 1

Reviewer 1 Report

Large cohort, and well decribed.

My most important comment is that it would be interesting to describe the consequences of this study for the field. It is obvious fibronectine measurements should be excluded from normal management. However what does CL add in the period after surgery? The consequence is a longer LOS, is this also a bias? Are there management changes that benefit these patients? It would be worth to elaborate on that and give recommendations. 

Author Response

First, we thank for your revision. We are glad to know that study is perceived as important.

Your comment raised is a relevant precision. We thus added a paragraph to the discussion:

Line 205-210: Taken together our results and the above considerations, independent factors such as CMS, PPROM and contractions definitely seem to play a more important role in prediction of preterm birth than preoperative CL measurements or perioperative ΔCL. The authors therefore conclude that neither preoperative CL measurements nor perioperative ΔCL, in the absence of other symptoms and unless CL measurement is < 95.P, must result in a different care management of patients that undergo or underwent fSB repair.

Reviewer 2 Report

1. The screening of a CL change post open fetal therapy with PTD outcome risk can be important in the management but no discussion is provided re possible management and although I recognize this was not a primary intent some discussion would benefit the reader.

2. Fetal FN is only useful with quantitative analysis as the single positive / negative result has no useful value. The null finding is useful to show don't waste the money. 

Author Response

‘1. The screening of a CL change post open fetal therapy with PTD outcome risk can be important in the management but no discussion is provided re possible management and although I recognize this was not a primary intent some discussion would benefit the reader.’

First, we thank for your revision. We are glad to know that study is perceived as important. Your comment raised is a relevant precision. We thus added a paragraph to the discussion:

Line 205-210: Taken together our results and the above considerations, independent factors such as CMS, PPROM and contractions definitely seem to play a more important role in prediction of preterm birth than preoperative CL measurements or perioperative ΔCL. The authors therefore conclude that neither preoperative CL measurements nor perioperative ΔCL, in the absence of other symptoms and unless CL measurement is < 95.P, must result in a different care management of patients that undergo or underwent fSB repair.

‘2. Fetal FN is only useful with quantitative analysis as the single positive / negative result has no useful value. The null finding is useful to show don't waste the money. ‘

We absolutely agree. To be more precise we added your comment on the quantitative analysis to the limitations of the study.

Line 234-236: No qfFN analysis was made and additionally, since we did not perform postoperative fFN testing, no statement regarding an eventual postoperative predictive validity can be made.

Round 2

Reviewer 1 Report

Good addition to the work